# The Effects of Different Feed Ration Levels on Growth, Welfare Rating, and Early Maturation in Juvenile Atlantic Salmon (*Salmo salar*)

Albert Kjartan Dagbjartarson Imsland [1,2,*] and Hjörtur Methúsalemsson [2,3]

1    Akvaplan-niva Iceland Office, Akralind 6, 201 Kópavogur, Iceland
2    Department of Biological Sciences, University of Bergen, High Technology Centre, 5020 Bergen, Norway; hjorturmethusalems@gmail.com
3    Arnarlax, Strandgata 1, 465 Bíldudalur, Iceland
*    Correspondence: albert.imsland@akvaplan.niva.no or albert.imsland@uib.no

**Abstract:** To investigate the possible effect of different feed ration levels on the growth, welfare, and early maturation of juvenile Atlantic salmon, 450 salmon parr with a mean (±standard error) initial weight of 51.6 g (±0.8) were reared in triplicate under three different feed ration levels for five months. The control group (100r) was fed every day, the 50r group was fed every other day, and the 33r group was fed every third day. In every group, 75 fish (half of the group) were individually tagged for monitoring of growth. The number of fin wounds was used as the welfare indicator, and to inspect the development of maturation, all fish were euthanized, and development of the gonads was monitored by visual inspection at the termination of the trial. The control group (100r) showed a significantly higher specific growth rate (0.90% day$^{-1}$) compared to the lower fed groups (50r, 0.67% day$^{-1}$ and 33r, 0.49% day$^{-1}$); however, the growth difference was 21–24% less than expected solely on the difference in the amount of feed given to each group. The 100r group showed the highest welfare rating, and the 33r group the lowest possible, indicating more aggressive behaviour and fin biting due to feed restriction in the 33r group. No difference ($p > 0.45$) was found in the development of maturation in females, but the combined numbers of males in maturity stages 2–5 showed an overall trend towards slower maturation in the 33r group compared to the 100r group. Although the present findings on the development of sexual maturation were subtle due to the limited time frame of the trial, the findings offer a foundation for future investigation into the relationship between the feed ration level and the development of sexual maturation in the rearing of juvenile Atlantic salmon.

**Keywords:** Atlantic salmon; feeding management; sexual maturation; fish welfare

**Key Contribution:** Reducing feeding during the rearing of juvenile Atlantic salmon leads to lower growth, increased fin damage, and reduced signs of early maturation in males.



## 1. Introduction

In the wild, Atlantic salmon (*Salmo salar*) face different periods where there is less access to food, which results in lack of access to feeding. Many different factors influence those periods such as changes in the environment or migration [1]. When animals experience a fasting period, three metabolic phases occur; first, they use stored glycogen, then they burn fat, and finally, muscle proteins are used [2]. Fish are ectotherms, with low metabolic rates, and can withstand a long fasting period (8–10 weeks) without suffering irreversible consequences [2]. Atlantic salmon have been reported to adapt to these changes by reducing their metabolic rate and using less energy to swim around, which helps the salmon to survive during these periods [3]. In the early life stages of the salmon, this has significant effects. Juvenile salmon rely heavily on obtaining all the energy needed to prepare them to grow and for smoltification. There is a powerful connection between surviving in the

ocean and the size of the fish during smoltification [4]. These changes can delay maturation since salmon require energy to undergo the process of becoming mature. Integral to the parr to smolt transformation and seawater adaptation are reductions in glycogen and changes in body lipids, including depletion of energy stores. Restricted feeding may lead to a disruption in the smoltification process, resulting in reduced hypo-osmoregulatory ability [5].

Farmed Atlantic salmon are starved over periods either voluntary or involuntary due to several factors. For example, feed withdrawal is carried out to (i) empty the gut of fish before they are handled (crowding, pumping, delousing, transportation, and harvest), (ii) starve fish that are suffering from a disease, or (iii) starve fish due to negative environmental conditions (temperature or hypoxia) [2]. Feeding control, by putting fish on starving periods, slows growth [6], and if the growth is reduced, it will affect the weight and size of the fish and could delay the timing of maturity [7]. In the aquaculture industry, if fish have insufficient access to feed, it results in slower growth and a longer production time. The fish will not be robust enough to prepare for smoltification and entering seawater. This can result in lower quality flesh, which affects the market value and the profitability of a company. It is very important for the managers of a company to monitor and manage feeding to promote the health and growth of their fish populations. Repeated/random periods of starvation can also have a negative effect on the welfare of fish.

Previous studies have investigated how fasting or reduced access to feed affects Atlantic salmon. Hvas et al. [8] investigated the effect of full starvation over four weeks, and the results showed that Atlantic salmon maintain their full swimming capacity as well as their ability to respond and recover from stress during an extended period of food deprivation [8]. In another trial Atlantic salmon weight, length, and condition factor did not change significantly during a fasting period of four weeks, and the fish immediately resumed eating upon refeeding. They concluded that starvation for up to four weeks has a minor effect on fish welfare [1]. The effect of different feed ration levels on juvenile Atlantic salmon have also been studied, and [9] found that food deprivation may result in significant osmotic disturbances in groups of juvenile Atlantic salmon fed a 0, 25, 50, 75, or 100% ration for six weeks and re-fed 100% after that and that the ration level significantly influenced the growth rate and mean body size. Martinez et al. [10] found that reducing the feed ration to 67% in juvenile Atlantic salmon (initial weight 23 g) did not help reduce maturation without significantly affecting growth [10].

Fish welfare is an important factor in modern aquaculture [11,12]. For welfare to have real meaning, the animal concerned must have the capacity for suffering [13,14], and recent evidence suggests that external stressors and painful stimuli cause avoidance in fish [13,14]. The welfare indicators presently used vary and may include, e.g., fin damage, the morbidity rate, and the mortality rate [15,16]; these indicators are commonly used/have been validated in farms. There are many factors that may cause increased fin damage and fin wounds in a rearing cage, with the feed ration level being one of these [17]. Aggression, as one form of social interaction, has the potential to cause physical injury. Among salmonid fish, aggression has evolved as a behavioural strategy. It is used in the wild to obtain and defend territories, to gain preferential access to food, and to maintain exclusive access to mates [15,17,18]. Monitoring and rating of fin damage is presently used as a welfare indicator in Icelandic salmonid culture [Kári Heiðar Árnason, Head of research station, Hólar University, Iceland, personal communication] and was, therefore, monitored in the present study.

The aim of this study was to investigate the effects of different feed ration levels on the growth, welfare rating, and maturation development of Atlantic salmon juveniles. Specifically, this study assessed the impact of varying feed ration levels on growth rates and welfare rating (here, measured as development of fin wounds) in juveniles. By examining the relationship between the feed ration level and the development of maturation in parr, this study aimed to provide insights into optimal feeding practices for the production of healthy Atlantic salmon juveniles.

## 2. Materials and Methods

### 2.1. Experimental Fish

Juvenile Atlantic salmon of the Saga strain from Benchmark genetics were obtained from the aquaculture company Arctic Sea Farm at Tálknafjörður (the Westfjords, Iceland). In April 2022, the fish were delivered to the Hólar University fish research station in Sauðárkrókur, Iceland, where they were kept and reared until the start of the experiment. The fish were reared in a flow-through system with an average ($\pm$SEM) temperature of 10 °C ($\pm$0.2), and the fish were fed ECO 3.0 feed (Table 1), which is manufactured by Laxá (Akureyri, Iceland).

**Table 1.** Overview of the content in ECO 3.0 feed used during the experimental period.

| | | |
|---|---|---|
| Content % | Protein | 49 |
| | Fat | 23 |
| | Carbohydrate | 13 |
| | Ash | 8 |
| | Dry material | 93 |
| | Panaferd, mg/kg | 50 |
| | Digestible energy (MJ/kg) | 19 |
| | Gross energy (MJ/kg) | 22.2 |
| Vitamins in kg of feed | Vitamin A IU | 2500 |
| | Vitamin D3 IU | 1500 |
| | Vitamin C (mg/kg) | 250 |
| | Vitamin E (mg/kg) | 115 |

### 2.2. Experimental Design and Sampling

The juvenile salmon were distributed randomly among 12, 2 m$^3$ tanks on the day of arrival from Tálknafjörður in April 2022. Every tank was provided with a steady stream of fresh water of approximately 10 °C. The oxygen saturation was kept above 83% (average $\pm$ SEM, 105 $\pm$ 5%) in all tanks throughout the experiment. The fish were exposed to continuous light (LD24:0) through the whole experiment.

On 13 August 2022, the fish were measured both by length and weight, and 450 juvenile salmon with no deformations and no visible wounds were distributed equally among nine 2 m$^3$ tanks (i.e., N = 50 in each tank). The initial stocking density was between 1.25 to 1.33 kg/m$^2$ in the nine tanks. Half of the fish in each tank were individually tagged with a Trovan® (Lifetest Vet Equipment, Hvidovrevej 80e, 2610 Rødovre, Denmark) Passive Integrate Transporter (PIT-tag, N = 225). The nine tanks were set up as three sperate lines. In each line, the feeding was reduced in two of the three tanks over the whole experimental period. One tank in each line had a ration of feed of 100r (feed every day, Control group), as used by the industry, one tank had a ration of feed of 50r (every other day), and the third one had a ration of feed of 33r (feed every third day). The fish were measured four times over the period of the experiment, from 13 August 2022 to 16 January 2023. If a fish was not in an optimal state, had many wounds, and was clearly unhealthy and exhibiting a lack of welfare (very small, open wounds with fin rays out), it was euthanized in a humane way. In those cases, the fish were placed in a 40-l tank, which included an overdose of anaesthesia (phenoxyethanol, 8–10 mL, Mjöll Frigg, Reykjavík, Iceland), and the fish were then euthanized via blunt force trauma to the head, and the gills were cut.

The fish were sampled on 13 August, 23 November, 13 December 2022, and 16 January 2023. Every fish in each tank was scanned for the PIT-tag, and the PIT-tag number, length (to nearest mm), and weight (to nearest 0.1 g) of the fish were measured and registered. The fish were also rated by welfare, from 0–5, according to how many wounds were observed [Kári Heiðar Árnason, Head of research station, Hólar University, Iceland, personal communication], with 0 as the best result and 5 as the worst. This scale was based on the percentage of wounds found on the eight fins of Atlantic salmon, i.e., 0 = 0–17% fins with wounds, 1 = 18–34%, 2 = 35–51%, 3 = 52–68%, 4 = 69–85%, and 5 $\geq$ 85%.

Fish were sampled for analyses of development of maturity on 13 December 2022 and on 16 January 2023. On 12 December, an accident occurred, where one of the tanks had an air bubble blocking the intake of water. All the fish in the tank died (N = 48), but as this happened one day prior to the planned sampling day, it was possible to rate the maturity of all the fish and include the data in the study. In addition, on 13 December 8, fish were sampled from each of the remaining 8 tanks (N = 64), euthanized with a blow to the head, cut open, and the development of maturity was rated according to [19] (Figure 1). In the final sampling on 16 January, all the fish (tagged and untagged) were euthanized, cut open, and the development of maturity was investigated.

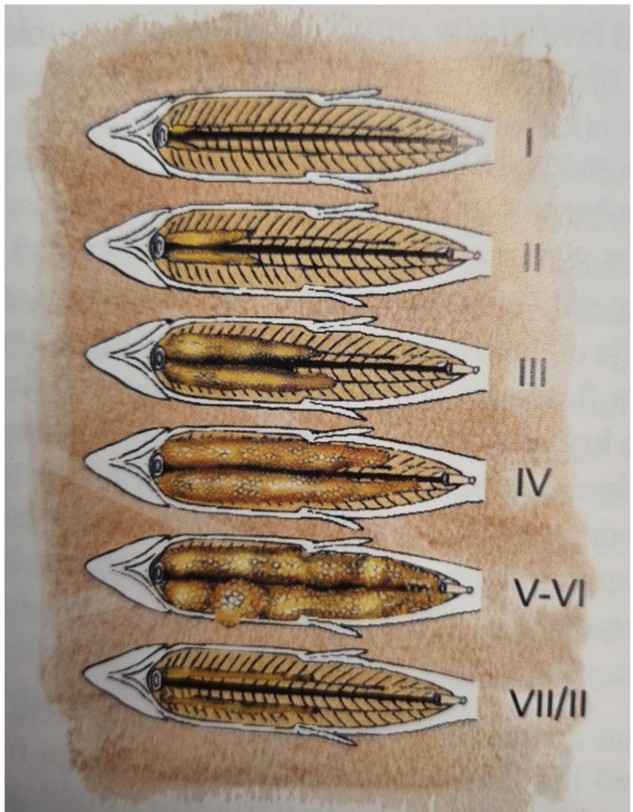

**Figure 1.** Scale for stages of maturity of Atlantic salmon. Modified from [18].

Due to signs of wounds and some fungal growth after the November sampling, it was decided to treat the fish with formalin bathing. Formalin bathing was conducted on 30 November 2022, where all the fish were treated with formalin at a ratio of 1:4000. The fish were starved one day prior to the treatment.

The condition factor (K) was calculated between sampling and from first day of the experiment until the last using Fulton's condition factor formula [20]:

$$\text{Condition Factor (K)} = 100 \, W/L^3$$

where W is the fish weight (g) and L is the fish length (cm).

The specific growth rate (SGR) was calculated between samplings, and for the whole period of the experiment, SGR was determined using the following formula [21]:

$$\text{SGR} = (\text{Ln} \, (W_t) - \text{Ln}(W_0)) \times 100/t(d)$$

where $W_0$ is the weight (g) at the beginning of the period, $W_t$ is the weight at the end of the period, and $t(d)$ is the length of the period in days.

### 2.3. Statistical Methods

All statistical analysis on the collected data was performed by using Microsoft Excel v. 365 (Microsoft Corporation, Redmond, WA, USA), and all data presented are the mean ± SEM unless otherwise specified. The distribution of response variables (body weight, body length, K, and SGR) were checked for normality and homogeneity of variance using the Kolmogorov-Smirnov [22] and Levene [23] test, respectively. To test for a possible difference in response variables, a two-way nested ANOVA [22] was performed. In cases of significant differences, a Bonferroni correction post hoc test was conducted based on each of the two-way models to identify where significant differences between groups occurred. Possible differences in mortality, maturity proportions, and welfare ratings between the experimental groups were tested with a $c^2$ test [22]. A significance level of $\alpha = 0.05$ was used in all cases if not stated otherwise.

### 2.4. Ethical Statement

Hólar University fish research station in Sauðárkrókur is approved as a laboratory animal department, and all handling of the experimental fish was carried out according to the station's ethical guidelines.

## 3. Results

### 3.1. Growth and Mortality

The average (±SEM) mortality was very low in all the groups (2.7% ± 0.5, in the 100r group, 4.0% ± 0.7 in the 50r group, and 2.0% ± 0.4 in the 33r group), and there was no significant difference in mortality between the groups (chi-squared test $p > 0.25$).

There was a significant (two-way nested ANOVA, $p < 0.01$) difference (Bonferroni post hoc test, $p < 0.01$) between the 33r group and the other groups in the mean ± SEM body weight and length (100r = 49.97 ± 0.81 g and 15.76 ± 0.08 cm, 50r = 50.67 ± 0.79 g and 15.80 ± 0.08 cm, and 33r = 53.42 ± 0.72 g and 16.03 ± 0.07 cm) at the start of the experiment. However, significant differences (two-way nested ANOVA, $p < 0.001$) in body weight (Figure 2A) and body length (Figure 2B) were found between the three experimental groups throughout the trial period. From November onwards, the fish in the 100r group were bigger compared to the two other experimental groups. The fish in the 33r group were the smallest throughout the trial period.

The condition factor (K) was not significantly different (two-way nested ANOVA, $p > 0.10$, Figure 3) between the experimental groups at the start of the experiment. From November onwards, the K was significantly higher in the 100r group compared to the two other groups. No difference in K between the 50r and 33r groups was found ($p > 0.15$, Figure 3). The K declined (two-way nested ANOVA, $p < 0.01$) in all three groups from August to November but was stable after that (Figure 3).

Apart from the period between November and December, a significant difference in the SGR (two-way nested ANOVA, $p < 0.001$, Figure 4) was found between all three experimental groups. The overall SGR for the whole experimental period differed significantly between the experimental groups (two-way nested ANOVA, $p < 0.001$, Figure 4). All groups had a steady SGR over the whole experiment, with individuals in the 100r group displaying a higher SGR than those reared at the lower feed ration levels of 50r and 33r (Bonferroni post hoc test, $p < 0.001$, Figure 4). Also, the fish in the 50r group displayed a significantly higher SGR than the fish in the 33r group. Overall, the SGR was 24% and 46% lower in the 50r and 33r groups, respectively, compared to the 100r group.

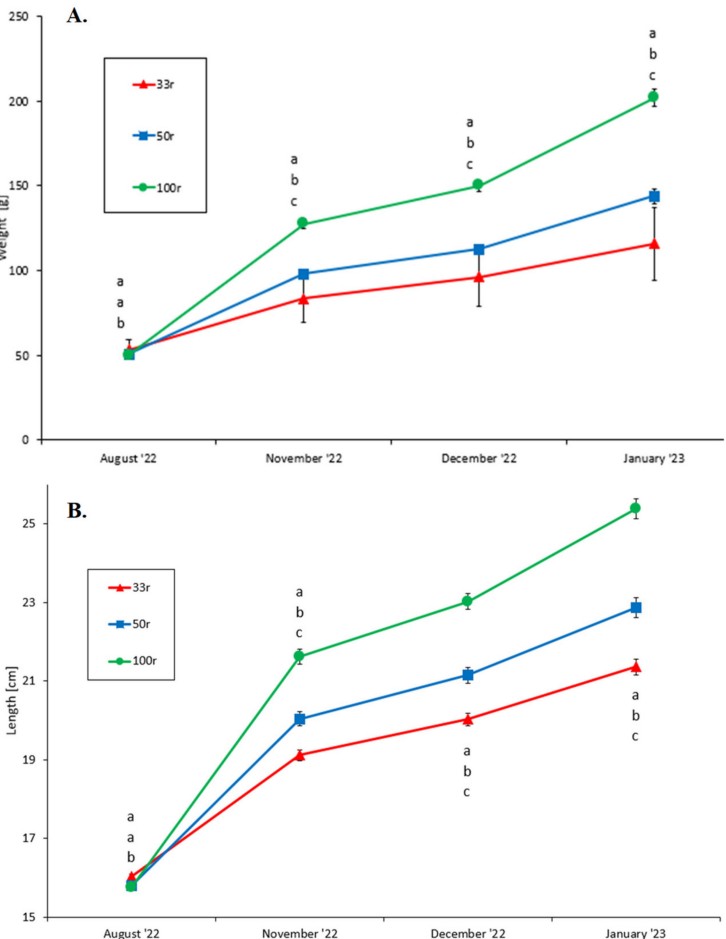

**Figure 2.** Mean weight (**A**) and length (**B**) of juvenile Atlantic salmon reared at different feed ration levels (100r, 50r, and 33r). Vertical lines indicate the SEM. Different letters indicate a statistical difference (Bonferroni post hoc test, $p < 0.05$) between the treatment groups at every sampling point. Note that (**B**) starts at 15 cm (indicated by a break in the Y-axis).

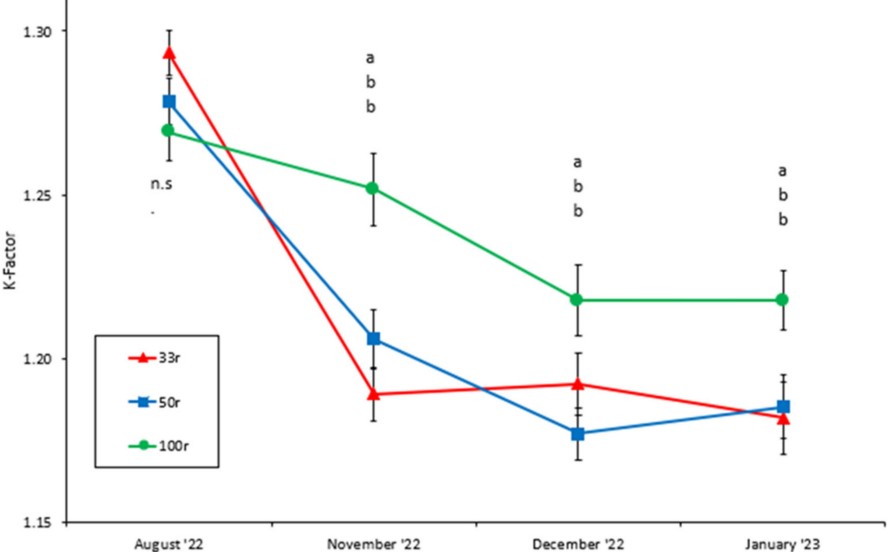

**Figure 3.** Mean condition factor of juvenile Atlantic salmon reared at different feed ration levels (100r, 50r, and 33r). Vertical lines indicate the SEM. Different letters indicate a statistical difference (Bonferroni post hoc test, $p < 0.05$) between the treatment groups at every sampling point. n.s. = no significant difference.

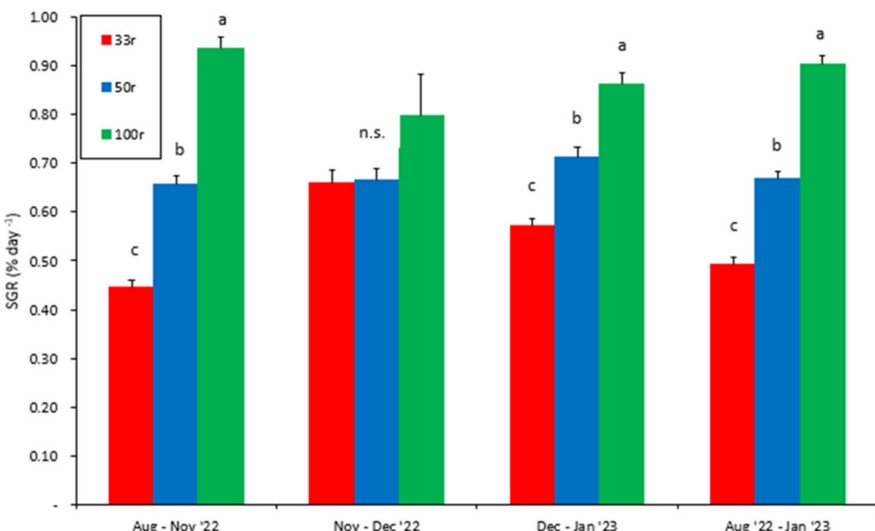

**Figure 4.** Mean SGR of juvenile Atlantic salmon reared at different feed ration levels (100r, 50r, and 33r). Vertical lines display the SEM. Different letters indicate a statistical difference (Bonferroni post hoc test, $p < 0.05$) between the treatment groups at every sampling point. n.s. = no significant difference.

### 3.2. Welfare Rating

There was a significant difference in the welfare ratings of the Atlantic salmon juveniles fed different feed ration levels in December ($c^2$ test, $p < 0.01$, Table 2) and January ($c^2$ test, $p < 0.01$, Table 3). Overall, the welfare rating was better for the 100r group than the other groups, and a better rating for the 50r group than for the 33r group was found. In December, the welfare rating differed between the 100r group and the two other groups for all welfare ratings except rating 4 and 5 ($c^2$ test, $p < 0.01$, Table 2). No differences were observed between the 50r and the 33r groups in December ($c^2$ test, $p > 0.25$, Table 2). In January (Table 3), similar group differences were observed as in December, with the overall better welfare rating in the 100r group, whereas differences were also observed between the 50r and the 33r groups for welfare ratings of 0, 3, and 5 ($c^2$ test, $p < 0.05$, Table 3).

**Table 2.** Frequency table of each welfare rating of juvenile Atlantic salmon reared at different feed ration levels (100r, 50r, and 33r) in December 2022. Superscript letters indicate significant differences between the experimental groups with [a] as the highest value.

| Welfare Rating | 100r | 50r | 33r | Total |
|:---:|:---:|:---:|:---:|:---:|
| 0 | 75 [a] | 32 [b] | 27 [b] | 134 |
| 1 | 48 [b] | 62 [a] | 65 [a] | 175 |
| 2 | 16 [b] | 29 [a] | 30 [a] | 75 |
| 3 | 8 [b] | 15 [a] | 21 [a] | 44 |
| 4 | 2 | 3 | 4 | 9 |
| 5 | 0 | 0 | 1 | 1 |
| Total | 149 | 141 | 148 | 438 |

### 3.3. Sexual Maturation

Overall, there was a minor connection between the feed ration level and the development of maturation in both sexes in the present study (Tables 4 and 5). There was a significant difference between the feeding ration groups and between the maturity in males in stage IV ($c^2$ test, $p < 0.05$, Table 4), but no significant difference in females ($c^2$ test, $p > 0.45$, Table 5). Further, there was a significant difference between the feed ration levels of

33r and 50r regarding males in maturity stage 0 ($c^2$ test, $p < 0.05$, Table 4). When looking at the combined numbers of males in stages II-V (Table 4), there was an overall trend towards lower numbers in the 33r group ($c^2$ test, $p = 0.06$) compared to the 100r group.

**Table 3.** Frequency table of each welfare rating of juvenile Atlantic salmon reared at different feed ration levels (100r, 50r, and 33r) in January 2023. Superscript letters indicate significant differences between the experimental groups with [a] as the highest value.

| Welfare Rating | 100r | 50r | 33r | Total |
|---|---|---|---|---|
| 0 | 35 [a] | 18 [b] | 5 [c] | 58 |
| 1 | 70 [a] | 51 [b] | 24 [b] | 145 |
| 2 | 11 [b] | 38 [a] | 28 [a] | 77 |
| 3 | 6 [c] | 12 [b] | 19 [a] | 37 |
| 4 | 0 [b] | 1 [b] | 5 [a] | 6 |
| 5 | 1 | 0 | 2 | 3 |
| Total | 123 | 120 | 83 | 326 |

**Table 4.** Frequency table of each maturation stage of male juvenile Atlantic salmon reared at different feed ration levels (100r, 50r, and 33r) in December 2022 (N = 52) and January 2023 (N = 215). Superscript letters indicate significant differences between the experimental groups with [a] as the highest value. The numbers within the brackets show results from the December sampling.

| Maturity Stage (0–V) | 100r-M | 50r-M | 33r-M | Total-M |
|---|---|---|---|---|
| 0 | 2 [ab] | 1 [b] | 3 [a] | 6 |
| I | 59 (11) | 59 (8) | 70 (28) | 188 (47) |
| II | 3 | 1 | 0 | 4 |
| III | 4 (3) | 2 | 4 (2) | 11 (5) |
| IV | 1 [b] | 3 [a] | 0 [b] | 4 |
| V | 2 | 0 | 0 | 2 |
| Total | 71 (14) | 66 (8) | 78 (30) | 215 (52) |

**Table 5.** Frequency table of each maturation stage of female juvenile Atlantic salmon reared at different feed ration levels (100r, 50r, and 33r) in December 2022 (N = 110) and January 2023 (N = 223). The numbers within brackets show results from the December sampling.

| Maturity Stage (0–V) | 100r-F | 50r-F | 33r-F | Total-F |
|---|---|---|---|---|
| 0 | 1 | 0 | 0 | 1 |
| I | 71 (10) | 78 (16) | 69 (34) | 219 (60) |
| II | 1 | 0 | 1 | 2 |
| III | 0 | 0 | 0 | 0 |
| IV | 0 | 0 | 0 | 0 |
| V | 1 | 0 | 0 | 1 |
| Total | 75 (10) | 78 (16) | 70 (34) | 223 (60) |

## 4. Discussion

As expected [6,9], the growth, length, condition factor (K), and specific growth rate results were increased when the feed ration level was 100r compared to the other feed ration levels. The differences between the 50r and 33r groups were more subtle. An initial decline in the K was seen in all groups, which is expected since it is characteristic

during smoltification [24]. However, although the fish were fed with different ration levels, surprisingly, the difference in mean weight and length did not match those percentages, e.g., the fish in the 50r group had only a 30% and 10% lower mean weight and length, respectively, compared to the fish in the 100r group. The same can be said when measuring the 50r group and the 33r group, an assumption of the difference would be that the 33r group would have the results of being 33% smaller than the 50r group, but the results showed that the fish in the 33r group had a 20% and 7% lower mean weight and length, respectively, compared to the fish in the 50r group. One possible explanation is that the juvenile salmon that had less feed utilised the feed better and slowed down other metabolic rates. In nature, Atlantic salmon face periods where there is less access to food, which results in lack of access to feeding. Cooke et al. [3] reported that salmon can adapt to changes by reducing their metabolic rate and, for example, using less energy to swim helps in survival during harsh periods [3]. But the current trial did not measure feed utilization, so it is not possible to verify this in the present study, although the data make this explanation seem plausible.

There is no universal definition for how to assess the welfare status of fish, and no consensus has been reached on definitions or assessment methodology [25]. According to [16], welfare outcome indicators, such as fin damage, the morbidity rate, and the mortality rate, should be used in standards and laws relating to salmon welfare. In the present trial, the welfare rating was decided according to the fin condition or amount of fin damage since fin damage is increasingly being used as a potential indicator of the welfare of farmed fish [17,25]. There were significant differences in the welfare ratings between all the feed ration groups, where the 100r group was rated with the best welfare rating, followed by the 50r group, and then the 33r group. There was also a progressive increase in fin damage in the 33r group possibly linked to increased aggressiveness during the last month of the trial. This supports that fin damage may reflect aggressive behaviour within the rearing unit [17,25] caused by the scarceness of food in this group. There are many indicators that may cause increased fin wounds on salmon, with the feeding ration level being one of these [17]. Overall, there was a very good relationship between the feeding ration level and the increase in fin wounds in the present trial supporting the idea that a reduction in the offered feed may lead to more aggressive behaviour in the fish, e.g., fin biting, thereby effecting the measured welfare rating.

Damage to the fins of salmonids is also caused by chronic infection with biofilm forming bacteria that progressively necrotize the fin edges [25]. Poor fin condition is coupled with a high stocking density, poor water quality, decreased condition factor, and increased cortisol levels or plasma glucose [25,26]. However, in this trial, there were no measurements of cortisol levels or plasma glucose, so it is not possible to deduce the possible causal relationship in relation to these variables. Water quality did not differ between the feeding groups, and the stocking density was low (<5 kg m$^{-3}$) in all groups. Density was not high in the tanks during the research, or at a maximum 5 kg per cubic meter. Fish stocking densities have been implicated in the occurrence of fin damage in Atlantic salmon. Higher fin damage has been described at high fish stocking densities [11,25,27]. However, as [17] stated, high fin damage can be observed at low fish stocking densities in Atlantic salmon in hatcheries as well.

As the fish in this study were small, it was expected to find only minor differences in maturation development in the fish. However, reduced growth can lead to less energy stores, e.g., lipid stores, needed for the sexual maturation process [7]. Such reduced growth and lowered energy stores can in theory delay the maturation of fish [7]. But, in the present trial, only minor differences in maturation development were found between the different feed ration levels among the juvenile Atlantic salmon. This could be traced back to that the experimental period was quite short (approx. 5 months), from August to January, and the fact that the fish in all groups were small (max 332.7 g). During the present trial period, only minor connections between the feed ration level and the development of maturation in all the juveniles were found, and the minor connection was only found in males; in

addition, no female surpassed maturity stage 2. The subtle findings found were for the earliest stage (0), where the 100r group differed from the 33r group, and the later stages (4), where the 100r group differed from the 50r group. This is in line with what [10] states that early maturation occurs mostly in males due to the lower energetic investments required for testis development in comparison to female egg production [10]. It is suggested that the relatively low maturity found in the males could be related to the short trial time, few fish, and that the fish were still relatively small and had not started full development of maturation. However, when the maturity stages of the males from 2–5 (since 0 and 1 rating equal barely started development of gonads) was combined, there was almost a significant difference between the 100r group and the 33r group. This difference, although subtle, might have been further enhanced if the trial period had been longer. Two precocious males (in stage 5) were found in the present trial, both found in the 100r group and both small (84.9 and 106.5 g). That could have caused a small deviation in the present trial since precocious male development is not directly linked to feeding but is also related to genetics [10].

## 5. Conclusions

The control group (100r) showed better growth and welfare overall compared to the lower fed groups (the 50r and the 33r). A small connection was found between different feed ration levels and early visual signs of maturation. Although findings were subtle due to the limited time frame of the trial, the findings offer a foundation for future investigation into the relationship between the feed ration level and the development of sexual maturation in the rearing of juvenile Atlantic salmon. This knowledge when further explored may help salmon producers with their smolt strategies and how they rear their juveniles and post-smolts.

**Author Contributions:** A.K.D.I. Conceptualization, Supervision, Writing—original draft, Review, and Editing; H.M. Investigation, Conceptualization, Visualization, Writing—original draft, Review, and Editing. All authors have read and agreed to the published version of the manuscript.

**Funding:** Financial support was given by the Aquaculture Environmental Fund of the Icelandic Ministry of Food, Agriculture and Fisheries, and the Research Council of Norway (grant 309288, FREMAD).

**Institutional Review Board Statement:** The present trial was conducted under the general ethical approval for fish trials at Hólar University Collage (Approval code: H-201530; Approval date: 1 January 2015).

**Data Availability Statement:** Data are contained within the article.

**Acknowledgments:** This project was hosted by the Hólar University Collage, Sauðárkróki, Iceland. We would like to thank Ólafur Sigurgeirsson, Rakel Þorbjörnsdóttir, Steven Price, and Kári Heiðar Árnason for their help during the research trial period.

**Conflicts of Interest:** Author Albert Kjartan Dagbjartarson Imsland was employed by the company Akvaplan-niva Iceland Office. Author Hjörtur Methúsalemsson was employed by the company Arnarlax. There are no conflicts of interest. The authors received public grants to conduct the study and those are not related to their working place in any way.

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
