# Peer review of "The Effects of Different Feed Ration Levels on Growth, Welfare Rating, and Early Maturation in Juvenile Atlantic Salmon (Salmo salar)"

_fishes, doi:10.3390/fishes9020070_

Round 1

Reviewer 1 Report

Comments and Suggestions for Authors

In general, the information appears clear, and no significant errors were detected

Author Response

This reviewer commented: "In general, the information appears clear, and no significant errors were detected". We appreciate the positive assessment of the reviewer. As no suggestions for changes were made, we have not altered the content of the ms based on this.

Reviewer 2 Report

Comments and Suggestions for Authors

‘The effects of different feed ratio on growth, welfare rating and early maturation in juvenile Atlantic salmon’

This study investigates the impact of reduced feed ration on different important parameters. It is generally well written and straight forward.

I have mostly minor comments.

My main comment is the use of ‘ratio’, sometimes ‘ration’. I think it is best to use ‘ration’ instead of ‘ratio’, in the title and throughout the manuscript

Abstract

L11, add that it was done in triplicate

L13, you should mention that this 75 fish tagged correspond to half of the fish per treatment, as you mention in the material and methods. It seems also a bit strange to have tagged half of the fish, why not all the fish?

L16, maybe change ‘fully fed’ to ‘control’

L22, change ‘lower’ to ‘slower’

Introduction
L31, add the latin name (Salmo salar) in italic the first time Atlantic salmon is mentioned (also L10 of the abstract as well).

L34, shouldn’t it be ‘glycogen’ here, as glucose is stored in this form?
L35, add ‘fish are ectotherms, with low metabolic rates’, the fact that they are ectotherms explains this low metabolic rate

L36, maybe add some timeline, i.e number of days/weeks they can go on fasting?
L50, please rephrase the last part of the sentence

L47-59. You can also add a part here saying that repeated/random periods of starvation is also not good for the fish

L57, in addition to the economical impact, and the impact on welfare, it can also have even worse effects, i.e death

L68-70, you can develop a bit more regarding studies done in the same area as yours, so the reader can know what has been done/what your study brings in more. Add the group that were studied, the time period of previous experiments ect, to provide a context and show the added value of your study.

L74, you mention pain here, you should quote a work of Lynne Sneddon, as she is the leading expert in this topic for fish

L76, when you quote the 14-15 papers, you should say that those indicators mentioned there are commonly used/have been validated form farms

Material and methods

L97, add the SEM of the temperature. Also, you describe your system a bit later on, but here you should at least mention the type (i.e flow through I guess?

L99, Table 1, you can delete the spaces in the table to make it more concise like the other tables.

For the digestible energy, you can use ‘19’ instead of ’19.0’ since you don’t mention ‘.0’ for the other parameters

L103, add the SEM of the oxygen

L107, you should say it was 50 fish/tank?
L109, add the reference (brand, city country) of the pit tags
L111, you should add that the 100r group is the control. You should mention as well if it was the same beforehand, i.e where the fish used to this regimen before/before being transferred to your facility? What is the rationale of feeding them everyday, is it standard in the industry?

L117, harmonize with the rest of the text, abbreviate litre to ‘l’

L120-125, good description, but you must provide pictures of the wounds to illustrate/show the reader, and for future reference of the scale used; what is a 0, what is a 1,2,3,4,5 ? You add a Figure 1 showing the maturity stages illustration, which are more commonly known than the scale you developed for wounds: it is important to show it/illustrate it

L129 and L131, remove ‘total’ on ‘Ntotal’, just use ‘N’
L135, Figure 1. This figure is quite big, and the resolution maybe not optimal. Also, there are differences between the numbers here and your tables, here it is I, II, II, IV, V-VI and VII/II, but you use 0,1,2,3,4,5 in your text, it is not matching. You should harmonize it. Either add ‘0,1,2,3,4,5’ on the figure, or use the same roman numbers in your tables

L126-142, there are a lot of different sampling events and other things happening. It would help the reader if maybe you add a figure with a timeline, showing when the fish arrived, the acclimation period, the different samplings, the formalin treatment ect

L141, you can change ‘K-factor’ to ‘K’ and also in the rest of the text and figures/legends

L147, change ‘*’ to ‘x’

L151 add the city/country for Microsoft

L153, change ‘condition factor’ to ‘K’, ‘specific growth rate’ to ‘SGR’ as you already explained the abbreviation

Results

You present the mortality results buy groups, i.e for all the replicates?
I think it should be presented as mean of the three replicates +/- SD to show if there is a tank effect, and maybe redo the statistics, not with a Chi.square but ANOVA like all the other parameters?

 L167, before mentioning the results of the posthoc Bonferroni, you should mention the result of the ANOVA

L168, you should add the SEM of the weight and length for all the groups.

L171-172. There is a statistical difference at the start, but is it really relevant biologically? 50.7 vs 53.4. That’s why you should really add the SEM as I suggested earlier to really emphasize that it is nothing and they are all the same at the start, this ‘significant difference’ is just random/chance

Figures 2 and 3. Since you mention always 100r first in your text and it is also your control group, it should be first in the legends in the figure. Since you only show the positive error bars in Figure 4, maybe do the same for those as well.

Figure 2B, you should start at 0.

Consider changing Figure 3 to a bar chart, as you show SGR on a bar chart, it would also make sense to do the same for K. Change the Y axis legend to ‘K’ instead of ‘K-factor’. Consider starting the Y axis at 0, maybe add some breaks.Changing to a bar chart will also allow you to better show the statistical differences with time, here you only show the differences between treatment at each time point, but there is also a time effect, the K is different between August 22 and the rest at least for 505 and 33r, right? It should appear on the figure somehow, use asterix?

L179, rephrase. ‘K was not significantly different’

L188-191, please rephrase, avoid the passive form

Tables 2-5, harmonize the legend with the figures, for the figures you say ‘at different feeding ratio(n)s (100r, 505, 33r)’, and here you say ‘in the three experimental groups’

Change the order of the column as well, you should start with 100r in the first column, as it is your control and you always start to mention that one

L221, add ‘for both sexes’

Tables 4 and 5, you used ‘()’ for the December, does it means that for example for the males 33r, if you have ‘70(28)’ for stage 1, you had 70 in total for both December and January, i.e you had 42 in January, 28 for December, OR that you had 70 for January, 28 for December, so 98 in total? IT is a bit confusing.
Also, as mentioned earlier, you use stages from 0-5, in the figure 1 the scale used different roman signs. Please harmonize.

Also, you have ‘significant differences’ for males, stage 4, but it’s 0, 3 and 1 individuals only there… can it really be considered a big difference? Maybe instead of displaying 6 stages, you could regroup your fist in 3 main stages, i.e 0-1; 2-3;4-5?

Discussion

L238, remove the comas before and after ‘[6,9]’

L242 and elsewhere, I would avoid using ‘%’ for the ration

L257, rephrase, ‘there is no universal definition of welfare’ instead of ‘the problem […]’

L265-266, could maybe this be explain by the fact that they are closer to the smoltification?

L267, you can also add that here it is caused by the scarceness of food

L273, change ‘more often’ by ‘also’? I think the fin damage mostly has for primary source some kind of wound, due to self damage (i.e tank) or caused by a fish (bite) or human (net ect), then the other factor you mention will jump in and create i.e necrosis ect, but there is a physical damage at the start, in most cases

L276, reverse the order, first mention cortisol, then glucose

L279. This should be also in the material and methods, the stocking density.
Also, the density is quite low compared to real farm situations.
Was this density of 5 kg/m3 kept throughout the trial, or was it the initial density? What was the final density then? You should mention if there are differences in densities throughout the trial
L307. Where those two fish in the stage 5 the biggest ones? Please mention the weight as well.

Conclusion

L311, change ‘100% fed’ to ‘fed normally’

Author Response

Reviewer 2

This study investigates the impact of reduced feed ration on different important parameters. It is generally well written and straight forward.

I have mostly minor comments.

My main comment is the use of ‘ratio’, sometimes ‘ration’. I think it is best to use ‘ration’ instead of ‘ratio’, in the title and throughout the manuscript

  • We have changed ‘ratio’ to ‘ration’ throughout the manuscript as suggested.

Abstract

L11, add that it was done in triplicate

  •  

L13, you should mention that this 75 fish tagged correspond to half of the fish per treatment, as you mention in the material and methods. It seems also a bit strange to have tagged half of the fish, why not all the fish?

  • Information added.
  • This was due to logical reasons as we did not have enough tags to tag all the fish.

L16, maybe change ‘fully fed’ to ‘control’

  • Changed as suggested.

L22, change ‘lower’ to ‘slower’

  • Changed as suggested.

Introduction

L31, add the latin name (Salmo salar) in italic the first time Atlantic salmon is mentioned (also L10 of the abstract as well).

  • Latin name added.

L34, shouldn’t it be ‘glycogen’ here, as glucose is stored in this form?

  • We have changed this.

L35, add ‘fish are ectotherms, with low metabolic rates’, the fact that they are ectotherms explains this low metabolic rate

  • Added as suggested.

L36, maybe add some timeline, i.e number of days/weeks they can go on fasting?

  • Added as suggested. Recent research show that Atlantic salmon can be starved for up to eight weeks without negative long-term consequences.

L50, please rephrase the last part of the sentence

  • We have rephrased this sentence.

L47-59. You can also add a part here saying that repeated/random periods of starvation is also not good for the fish

  • Good suggestion that we have followed.

L57, in addition to the economical impact, and the impact on welfare, it can also have even worse effects, i.e death

  • We have added this.

L68-70, you can develop a bit more regarding studies done in the same area as yours, so the reader can know what has been done/what your study brings in more. Add the group that were studied, the time period of previous experiments ect, to provide a context and show the added value of your study.

  • We have rephrased this section in line with the reviewer suggestion.

L74, you mention pain here, you should quote a work of Lynne Sneddon, as she is the leading expert in this topic for fish

  • We have added citation to the work of Lynne Sneddon and colleagues.

L76, when you quote the 14-15 papers, you should say that those indicators mentioned there are commonly used/have been validated from farms

  • Changed as suggested.

Material and methods

L97, add the SEM of the temperature. Also, you describe your system a bit later on, but here you should at least mention the type (i.e flow through I guess?

  • SEM added as suggested and type of system (flow through) specified.

L99, Table 1, you can delete the spaces in the table to make it more concise like the other tables.

  • Changed as suggested.
  •  

For the digestible energy, you can use ‘19’ instead of ’19.0’ since you don’t mention ‘.0’ for the other parameters

  • Changed as suggested.

L103, add the SEM of the oxygen

  • SEM for oxygen added.

L107, you should say it was 50 fish/tank?

  • Yes, and this has been changed.

L109, add the reference (brand, city country) of the pit tags

  • More information about the pit tags added.

L111, you should add that the 100r group is the control. You should mention as well if it was the same beforehand, i.e where the fish used to this regimen before/before being transferred to your facility? What is the rationale of feeding them everyday, is it standard in the industry?

  • We have added the requested information.

L117, harmonize with the rest of the text, abbreviate litre to ‘l’

  • Harmonized as suggested.

L120-125, good description, but you must provide pictures of the wounds to illustrate/show the reader, and for future reference of the scale used; what is a 0, what is a 1,2,3,4,5 ? You add a Figure 1 showing the maturity stages illustration, which are more commonly known than the scale you developed for wounds: it is important to show it/illustrate it

  • Unfortunately, we don’t have pictures available to illustrate the use of this scale. We have, therefore, added a more detailed description of the scale in the text.

L129 and L131, remove ‘total’ on ‘Ntotal’, just use ‘N’

  • Changed as suggested.

L135, Figure 1. This figure is quite big, and the resolution maybe not optimal. Also, there are differences between the numbers here and your tables, here it is I, II, II, IV, V-VI and VII/II, but you use 0,1,2,3,4,5 in your text, it is not matching. You should harmonize it. Either add ‘0,1,2,3,4,5’ on the figure, or use the same roman numbers in your tables

  • We have changed the numbers to roman numbers as this was used in the reference study.

L126-142, there are a lot of different sampling events and other things happening. It would help the reader if maybe you add a figure with a timeline, showing when the fish arrived, the acclimation period, the different samplings, the formalin treatment ect

  • We have simplified the text to make the timeline clearer.

L141, you can change ‘K-factor’ to ‘K’ and also in the rest of the text and figures/legends

  • Changed as suggested.

L147, change ‘*’ to ‘x’

  • Changed as suggested.

L151 add the city/country for Microsoft

  • Information added.

L153, change ‘condition factor’ to ‘K’, ‘specific growth rate’ to ‘SGR’ as you already explained the abbreviation

  • Changed as suggested.

Results

You present the mortality results buy groups, i.e for all the replicates?

I think it should be presented as mean of the three replicates +/- SD to show if there is a tank effect, and maybe redo the statistics, not with a Chi.square but ANOVA like all the other parameters?

  • The mortality in the trial was very low and did not vary between tanks as 1-2 dead fish were found in all tanks during the whole experimental period. We see the reviewers point of adding the SEM and have added this information.

 L167, before mentioning the results of the posthoc Bonferroni, you should mention the result of the ANOVA

  • We have added the ANOVA information.

L168, you should add the SEM of the weight and length for all the groups.

  • We have added SEM information as suggested.

L171-172. There is a statistical difference at the start, but is it really relevant biologically? 50.7 vs 53.4. That’s why you should really add the SEM as I suggested earlier to really emphasize that it is nothing and they are all the same at the start, this ‘significant difference’ is just random/chance

  • We agree with the reviewer assessment that this minor difference (albeit significant) has little biological bearing. As suggested the SEM for all groups have been added showing that this difference is in fact minimal and could well be due to random effects.

Figures 2 and 3. Since you mention always 100r first in your text and it is also your control group, it should be first in the legends in the figure. Since you only show the positive error bars in Figure 4, maybe do the same for those as well.

  • By default, the 33r groups comes first in the legend (we presume this is linked to the set-up of the datafile). Hopefully, this is not to annoying. If needed we could remove the legends in the figures.
  • It is customary to show both positive and negative variation in line plots and only the positive ones in bar plots. We have just followed this customary here.

Figure 2B, you should start at 0.

  • We agree that it is preferable to start all figures at origo, but by doing so in this figure the subtle differences between the groups would be very difficult to grasp for the reader. We have therefore indicated that we don’t start the figure at origo by adding a break in the Y-axis.

Consider changing Figure 3 to a bar chart, as you show SGR on a bar chart, it would also make sense to do the same for K. Change the Y axis legend to ‘K’ instead of ‘K-factor’. Consider starting the Y axis at 0, maybe add some breaks. Changing to a bar chart will also allow you to better show the statistical differences with time, here you only show the differences between treatment at each time point, but there is also a time effect, the K is different between August 22 and the rest at least for 505 and 33r, right? It should appear on the figure somehow, use asterix?

  • As the K is directly linked to the mean weight and length is makes perfect sense to show the development in K in a line plot as for the mean weight and length.
  • We have amended the Y-axis legend as suggested.
  • The changes over time is already explained in the text (significant decline in all three groups between August and November).

L179, rephrase. ‘K was not significantly different’

  • Rephrased as suggested.

L188-191, please rephrase, avoid the passive form

  • Rephrased as suggested.

Tables 2-5, harmonize the legend with the figures, for the figures you say ‘at different feeding ratio(n)s (100r, 505, 33r)’, and here you say ‘in the three experimental groups’

  • Text harmonized as suggested. Many thanks for notifying this discrepancy.

Change the order of the column as well, you should start with 100r in the first column, as it is your control and you always start to mention that one

  • Changed as suggested.

L221, add ‘for both sexes’

  • Added as suggested.

Tables 4 and 5, you used ‘()’ for the December, does it means that for example for the males 33r, if you have ‘70(28)’ for stage 1, you had 70 in total for both December and January, i.e you had 42 in January, 28 for December, OR that you had 70 for January, 28 for December, so 98 in total? IT is a bit confusing.

  • We agree that this may appear a bit confusing, but we decided to show the data from both December and January in the same table (alternatively we could have divided both tables into two tables, but adding more length to the ms which is not desirable).
  • The data in brackets show the December sampling so it should read (for the male 33r group) 28 for December and 70 for January i.e. 98 in total.

Also, as mentioned earlier, you use stages from 0-5, in the figure 1 the scale used different roman signs. Please harmonize.

  • We have harmonized the table text in line with Figure 1.

Also, you have ‘significant differences’ for males, stage 4, but it’s 0, 3 and 1 individuals only there… can it really be considered a big difference? Maybe instead of displaying 6 stages, you could regroup your fist in 3 main stages, i.e 0-1; 2-3;4-5?

  • We agree, this difference is very subtle and bears little biological meaning. This is the reason we have combined stages II-V and tested for differences for those stages in the males.

Discussion

L238, remove the comas before and after ‘[6,9]’

  • Changed as suggested.

L242 and elsewhere, I would avoid using ‘%’ for the ration

  • We have changed this here and elsewhere in the ms.

L257, rephrase, ‘there is no universal definition of welfare’ instead of ‘the problem […]’

  • We have rephrased this sentence.

L265-266, could maybe this be explain by the fact that they are closer to the smoltification?

  • Yes, this may be part of the explanation but we think it has more do to with the long term absence of enough feed leading to more aggressive behaviour. This is something we noted during the daily observations of the experimental fish.

L267, you can also add that here it is caused by the scarceness of food

  • Yes, we agree and have added this.

L273, change ‘more often’ by ‘also’? I think the fin damage mostly has for primary source some kind of wound, due to self damage (i.e tank) or caused by a fish (bite) or human (net ect), then the other factor you mention will jump in and create i.e necrosis ect, but there is a physical damage at the start, in most cases

  • We fully agree and have changed this.

L276, reverse the order, first mention cortisol, then glucose

  • We have reversed the order.

L279. This should be also in the material and methods, the stocking density.

Also, the density is quite low compared to real farm situations.

Was this density of 5 kg/m3 kept throughout the trial, or was it the initial density? What was the final density then? You should mention if there are differences in densities throughout the trial

  • We have added the initial stocking density in the Materials and Methods (between 1.25 and 1.33 kg/m3).
  • The density of 5 kg/m3 was the final density and we agree that this is a very low density compared to real farm situations.

L307. Where those two fish in the stage 5 the biggest ones? Please mention the weight as well.

  • No, they were actually among the smallest ones (85 and 105 g). we have added this information.

Conclusion

L311, change ‘100% fed’ to ‘fed normally’

  • Changed as suggested.

Reviewer 3 Report

Comments and Suggestions for Authors

Please find in the attachment

Author Response

Reviewer 3

This reviewer submitted a marked ms with highlighted comments. We have carefully amended the revised ms in line with those comments. Changes are commented below:

  • Line 3. Scientific name has been added to title.
  • Line 10. SEM written in full.
  • Line 16. Significant difference.
  • Line 20, p value added.
  • Line 26. Apart from the species name the keywords don’t repeat words in the title.
  • Line 36. Length of fasting periods in previous trials added.
  • Materials and methods. Ethical statement added (new section 2.4).
  • Line 99. Yes, this is normal feed for juvenile Atlantic salmon.
  • Lines 310-316. We have added more on future investigations and possible use.
  • The references are as updated as possible. The old ones are part of the standard literature for Atlantic salmon and are therefore included.